# Proteomics and Bioinformatics Identify Drug-Resistant-Related Genes with Prognostic Potential in Cholangiocarcinoma

**DOI:** 10.3390/biom14080969

**Published:** 2024-08-08

**Authors:** Kankamol Kerdkumthong, Sittiruk Roytrakul, Kawinnath Songsurin, Kandawasri Pratummanee, Phanthipha Runsaeng, Sumalee Obchoei

**Affiliations:** 1Department of Biochemistry, Division of Health and Applied Sciences, Faculty of Science, Prince of Songkla University, Hat Yai District, Songkhla 90110, Thailand; kkerdkumthong@gmail.com (K.K.); kawinnath.ss@gmail.com (K.S.); kandawasri1122@gmail.com (K.P.); phanthipha.r@psu.ac.th (P.R.); 2Functional Proteomics Technology Laboratory, National Center for Genetic Engineering and Biotechnology, National Science and Technology Development Agency, Pathum Tani 12120, Thailand; sittiruk@biotec.or.th; 3Center of Excellence for Biochemistry, Faculty of Science, Prince of Songkla University, Hat Yai District, Songkhla 90110, Thailand

**Keywords:** cholangiocarcinoma, quantitative proteomics, drug resistance, 5-fluorouracil, gemcitabine, EMT

## Abstract

Drug resistance is a major challenge in the treatment of advanced cholangiocarcinoma (CCA). Understanding the mechanisms of drug resistance can aid in identifying novel prognostic biomarkers and therapeutic targets to improve treatment efficacy. This study established 5-fluorouracil- (5-FU) and gemcitabine-resistant CCA cell lines, KKU-213FR and KKU-213GR, and utilized comparative proteomics to identify differentially expressed proteins in drug-resistant cells compared to parental cells. Additionally, bioinformatics analyses were conducted to explore the biological and clinical significance of key proteins. The drug-resistant phenotypes of KKU-213FR and KKU-213GR cell lines were confirmed. In addition, these cells demonstrated increased migration and invasion abilities. Proteomics analysis identified 81 differentially expressed proteins in drug-resistant cells, primarily related to binding functions, biological regulation, and metabolic processes. Protein–protein interaction analysis revealed a highly interconnected network involving MET, LAMB1, ITGA3, NOTCH2, CDH2, and NDRG1. siRNA-mediated knockdown of these genes in drug-resistant cell lines attenuated cell migration and cell invasion abilities and increased sensitivity to 5-FU and gemcitabine. The mRNA expression of these genes is upregulated in CCA patient samples and is associated with poor prognosis in gastrointestinal cancers. Furthermore, the functions of these proteins are closely related to the epithelial–mesenchymal transition (EMT) pathway. These findings elucidate the potential molecular mechanisms underlying drug resistance and tumor progression in CCA, providing insights into potential therapeutic targets.

## 1. Introduction

Cholangiocarcinoma (CCA), a malignancy affecting the bile ducts, is relatively rare on a global scale. However, its prevalence and mortality rates are notably elevated in specific Asian regions, with the highest occurrences observed in Northeast Thailand [1]. Due to late-stage diagnosis, most CCA cases are ineligible for curative surgical interventions. Consequently, chemotherapy plays a crucial role in the treatment of these late-diagnosed CCA patients [2]. However, the effectiveness of chemotherapy is limited by the acquisition of drug resistance by CCA cells, leading to inevitable tumor recurrence [3]. A deeper understanding of the mechanisms underlying drug resistance holds the potential to enhance the effectiveness of CCA treatment. In the field of cancer research, to identify novel drug-resistant mechanisms or investigate the potency of novel treatments for combating drug resistance, drug-resistant cell lines are often employed [4,5,6,7].

Proteomics techniques have played a crucial role in advancing our understanding of cancer biology by investigating modifications in signaling pathways within tumor cells. This progress has led to the creation of comprehensive cancer proteome databases, merging molecular and clinical data, and identifying potential therapeutic targets and prognostic markers [8,9,10,11]. Recent examples of such revelations have been observed in various cancers including hepatocellular carcinoma [12], pancreatic cancer [13], glioblastoma [14], and lung adenocarcinoma [15]. However, to our knowledge, proteomic profiling for the elucidation of chemotherapeutic drug-resistant CCA cells has not yet been undertaken.

In the present study, we established 5-fluorouracil (5-FU) and gemcitabine-resistant cell lines named KKU-213FR and KKU-213GR, respectively, from the KKU-213A parental cell line. Various tumor characteristics such as the drug sensitivity, cell migration, and cell invasion capability of these cells were investigated. Furthermore, quantitative proteomic analysis was conducted to identify differentially expressed proteins in drug-resistant cell lines compared with the parental cell line and to explore the underlying drug-resistant mechanisms. Additionally, various bioinformatic tools were utilized to explore functions and protein–protein interaction networks. The Cancer Genome Atlas Program (TCGA) data were analyzed to investigate the clinical relevance of the key identified proteins.

## 2. Materials and Methods

### 2.1. Cell Culture

The KKU-213A, formerly known as M213, was established from liver fluke-associated intrahepatic mass-forming CCA tissue with an adenosquamous carcinoma nature from a Thai patient [16,17]. It was obtained from the Japanese Collection of Research Bioresources (JCRB) Cell Bank in Osaka, Japan. The cell line was cultured under standard conditions in a humidified atmosphere with 5% CO_2_ at 37 °C using Dulbecco’s Modified Eagle’s Medium (DMEM, Invitrogen, Carlsbad, CA, USA), supplemented with 25 mM glucose, 10% fetal bovine serum (FBS, Gibco, Paisley, Scotland), and 1% antibiotic-antimycotic mixture (Gibco, Grand Island, NY, USA).

### 2.2. Establishment of Drug-Resistant CCA Cell Lines

The chemotherapeutic drugs 5-FU and gemcitabine were purchased from Sigma (Sigma Aldrich, Saint Louis, MO, USA). Drug-resistant cell lines were established using a stepwise concentration incremental method, as previously described [18], with necessary modifications. The parental cell line of KKU-213A cells was initially cultured in DMEM containing the IC_25_ concentrations of 5-FU and gemcitabine. After four passages, the drug concentration was increased two-fold. Finally, cell lines growing exponentially in the presence of 7 µM 5-FU and 3 µM gemcitabine were designated as KKU-213FR and KKU-213GR, respectively.

### 2.3. Chemotherapeutic Drug Sensitivity Assay

The drug-resistant characteristic of the established KKU-213FR and KKU-213GR cell lines was verified through the 3-(4,5-dimethylthiazol-2-yl)-2,5-diphenyl-2H-tetrazolium bromide (MTT) assay. In this process, both drug-resistant cells and their parental counterparts were seeded at a density of 3 × 10^3^ cells/well in a 96-well plate and incubated for 24 h. Subsequently, they were exposed to varying concentrations of 5-FU or gemcitabine for 72 h. Afterward, an MTT solution at a concentration of 0.5 mg/mL was introduced into each well and incubated at 37 °C for 2 h. The resulting formazan crystals were dissolved by replacing the medium with 100 µL of dimethyl sulfoxide (DMSO). The absorbance was measured at 540 nm using a microplate reader, enabling the calculation of cell viability percentages for comparison between the drug-resistant and parental cell lines.

### 2.4. Cell Migration and Invasion Assay

The migration and invasion abilities of cells were evaluated using a modified Boyden chamber technique. In short, 2 × 10^4^ cells were placed onto an 8-µm porous polycarbonate membrane in the upper chamber. For the invasion assay, the membrane was pre-coated with 0.5 mg/mL Matrigel (Corning, Bedford, MA, USA). The lower chamber contained a complete medium to encourage cell migration in response to chemoattractants. After 12 h, any non-migrated cells in the upper chamber were removed and the membrane was fixed and stained using a 0.4% sulforhodamine B (SRB) solution. The numbers of migrated and invaded cells were then counted and compared across five randomly chosen low-power fields.

### 2.5. Proteomics Analysis Using LC-MS/MS

Cell pellets from KKU-213A, KKU-213FR, and KKU-213GR were lysed with 0.5% SDS and centrifuged at 10,000× *g* for 15 min. The resulting supernatant was transferred to a new tube, mixed with two volumes of cold acetone, and incubated overnight at −20 °C. The mixture was thawed and centrifuged again at 10,000× *g* for 15 min. Following the removal of the supernatant, the pellet was dried and stored at −80 °C until needed. Protein concentration of all samples was determined using the Lowry protein assay, with bovine serum albumin (BSA) as the standard protein. To reduce disulfide bonds, a solution of 10 mM dithiothreitol in 10 mM ammonium bicarbonate was added to the protein solution. To prevent reformation of disulfide bonds, the solution was treated with 30 mM iodoacetamide in 10 mM ammonium bicarbonate. Protein samples were digested with sequencing-grade porcine trypsin (at a ratio of 1:20) for 16 h at 37 °C. Subsequently, 100 ng of digested peptides were injected into an Ultimate3000 Nano/Capillary LC System (Thermo Scientific, Oxford, UK) coupled to an HCTUltra LC-MS system (Bruker Daltonics Ltd., Hamburg, Germany) equipped with a nano-captive spray ion source.

Briefly, the peptide digests were enriched on a μ-Precolumn (300 μm I.D. × 5 mm) C18 PepMap 100 with 5 μm, 100 Å (Thermo Scientific, UK). The peptides were then separated on a 75 μm I.D. × 15 cm column packed with Acclaim PepMap RSLC C18, 2 μm, 100 Å, nanoViper (Thermo Scientific, UK). This column was maintained in a thermostatted column oven at 60 °C. The column was supplied with solvents A and B, which contained 0.1% formic acid in water and 0.1% formic acid in 80% acetonitrile, respectively. Tryptic peptides were eluted from the column using a gradient of 5–55% solvent B at a constant flow rate of 0.30 μL/min for 30 min. Electrospray ionization was performed at 1.6 kV using the CaptiveSpray method with nitrogen as the drying gas at a flow rate of 50 L/h. Mass spectra (MS) and MS/MS spectra were acquired in the positive-ion mode at 2 Hz, covering a range of 150–2200 *m*/*z*. The collision energy was set at 10 eV based on the *m*/*z* value. MS/MS data-dependent acquisition was enabled. The ion trap was configured for positive-ion mode utilizing the manufacturer-specified standard enhanced mode with a scan rate of 8100 *m*/z/s. The scan range spanned from *m/z* 400 to 1500 for MS, averaging five spectra, and accumulated either 100,000 charges [by ion charge control (ICC)] or 200 ms, whichever came first. Collision-induced dissociation (CID) fragmentation was performed on the five most intense ions within the *m/z* range of 200–2800. The threshold for precursor ion selection was set at an absolute intensity of 5000. To prevent redundancy, strict active exclusion was applied: a precursor ion was excluded after two spectra and then reintroduced after a brief 0.3-min interval. The samples were run in three biological replicates of KKU-213A, KKU-213FR, and KKU-213GR, with each replicate undergoing LC-MS analysis in triplicate. A spike of digested BSA was used as an internal standard.

The DecyderMS 2.0 Differential Analysis software was employed to quantify peptides in individual samples from MS/MS data [19,20]. It provides novel 2D and 3D visualizations of LC-MS data to allow for raw data quality assessment and interactive confirmation of results obtained through automated methods for peptide detection, charge state assignments, and peptide matching across multiple LC-MS experiments. Univariate statistical tools such as Student’s *t*-test and ANOVA are available to identify significantly varying peptides among different groups of samples. Briefly, the raw LC-MS/MS data files were imported into the program, which transformed the information into a virtual peptide image. These images were then processed within the Pepdetect module of DecyderMS. The program initially identifies individual peptides by placing a box around them. Subsequently, ion counts for each peptide are integrated, and the Log_2_ of this number is reported. The process is repeated for each replicate sample. This resulting data are then uploaded to the Pepmatch module, where peptides within different virtual images are matched based on *m*/*z* and retention time. Notably, because the data was acquired in an ion trap mass spectrometer, tandem MS ensures accurate peptide matching (i.e., MS/MS spectra consistency). Within the Pepmatch module, each virtual image is assigned to a specific group (KKU-213A, KKU-213FR, and KKU-213GR). Group-to-group comparisons are performed. Integrated peptide counts serve as the basis for statistical analyses, assessing the probability that observed changes are not due to chance. False discovery rates are determined from the data.

MASCOT software, version 2.2 (Matrix Science, London, UK), was used to correlate MS/MS spectra obtained from DecyderMS software to the Homo sapiens protein database (downloaded on 18 August 2021). The search criteria included taxonomy (Homo sapiens), enzyme (trypsin), variable modifications (oxidation of methionine residues), mass values (monoisotopic), protein mass (unrestricted), peptide mass tolerance (1.0 Da), fragment mass tolerance (±0.4 Da), peptide charge states (1+, 2+, and 3+), and a maximum number of missed cleavages [21]. Proteins were identified using one or more peptides with an individual MASCOT score corresponding to *p* < 0.05 and subsequently annotated by UniProtKB/Swiss-Prot entries (http://www.uniprot.org/; accessed on 18 August 2021).

### 2.6. Identification of Differentially Expressed Proteins

MetaboAnalyst 6.0 was used for the statistical evaluation of the identified proteins [22]. To illustrate the distinction between various groups of identified proteins, partial least squares-discriminant analysis (PLS-DA) was utilized. Differentially expressed proteins (DEPs) were selected if they showed at least a 3-fold significant difference between drug-resistant cell lines and the parental line, with a false discovery rate (FDR) < 0.05.

Venn diagrams were generated to show the total number of upregulated and downregulated proteins. To visualize and compare the Log2 expression values of all DEPs, a heatmap was generated using MetaboAnalyst 6.0 [22].

### 2.7. Bioinformatics Analysis

Protein organization and biological action were conducted on differentially expressed proteins using the protein analysis through evolutionary relationships (Panther software version 18.0; http://pantherdb.org/; accessed on 5 December 2022) protein classification [23]. The categorization included molecular function, biological process, and protein class.

The gene symbols of all upregulated proteins sharing in both KKU-213FR and KKU-213GR were queried against TCGA data through The University of Alabama at Birmingham Cancer data analysis portal (UALCAN) (http://ualcan.path.uab.edu/index.html; accessed on 12 July 2023) [24] to analyze their mRNA expression levels in CCA patients’ tissues. The genes that were upregulated in both drug-resistant cell lines, as indicated by proteomics results, and whose mRNA expression levels were significantly elevated in CCA patients’ tissues according to TCGA data, were filtered and presented in a heatmap. These genes were then subjected to protein–protein interaction prediction and generation using STRING version 11.5 (https://version-11-5.string-db.org; accessed on 30 July 2023) [25]. The following parameters were used: Organism: Homo sapiens; Network type: Full STRING network; Required score: medium confidence (0.4); and FDR stringency: medium (5 percent). The largest protein network, comprising six proteins, was assigned as the ‘focused network’ and further analyzed. The mRNA expression of the six selected genes and the correlations among them were assessed through expression and correlation analysis using the Gene Expression Profiling Interactive Analysis 2 (GEPIA2) platform (http://gepia2.cancer-pku.cn/#correlation; accessed on 7 August 2023). The dataset of cholangiocarcinoma (CHOL) and gastrointestinal tract (GI) cancer patients was utilized. The correlation between genes was computed with Pearson correlation coefficients and presented in scatter plots.

### 2.8. Survival Analysis

The prognostic values of the six genes within the focus network were assessed using TCGA data through the GEPIA2 web portal. Overall survival (OS) and disease-free survival (DFS) were analyzed through Kaplan–Meier survival plots and heatmaps (http://gepia2.cancer-pku.cn/#survival; accessed on 7 August 2023). Heatmap analysis was employed to compare the contributions of genes to survival in gastrointestinal (GI) tract cancers, with estimates generated using the Mantel–Cox test at a significance level of less than 0.05. Survival analysis was performed based on the expression status of genes in the focused network, and a Kaplan–Meier curve was plotted. The Cox proportional hazards model, including a 95% confidence interval (CI), was used to evaluate the significance of genes in predicting OS and DFS.

### 2.9. RT-qPCR

The mRNA levels of the six chosen genes were assessed using RT-qPCR, with the beta-actin gene (ACTB) serving as an internal control. Initially, total RNA was isolated from KKU-213A, KKU-213FR, and KKU-213GR cells, as well as from the drug-resistant cells transfected with siRNAs, using the GF-1 total RNA extraction kit (Vivantis, Malaysia), and then converted into cDNA using the RevertAid First Strand cDNA Synthesis Kit (Thermo Scientific, Vilnius, Lithuania). The resulting cDNA was amplified using gene-specific primers (Table 1) and Maxima SYBR Green/ROX qPCR master Mix (Thermo Scientific, Vilnius, Lithuania), following the manufacturer’s instructions. The qPCR analysis was conducted on an Agilent Technologies Stratagene Mx3005P instrument. The relative mRNA expression levels were normalized using the internal control gene and calculated using the 2^−ΔΔCt^ method.

### 2.10. Gene Knockdown Using siRNA

Specific siRNA sequences targeting MET, LAMB1, ITGA3, NOTCH2, CDH2, and NDRG1 mRNA, as well as a non-targeting negative control siRNA (siNC), were obtained from Gene Universal Inc. (Newark, DE, USA) and are shown in Table 2. Briefly, for gene knockdown, 2 × 10^5^ cells per well were plated in a six-well plate overnight. The next day, the cells were washed and serum-starved in 800 µL per well of Opti-MEM^®^ (Gibco, Grand Island, NY, USA) for 1 h. Then, 200 µL of transfection complex containing 100 pmoles siRNA and 2 µL Lipofectamine™ 2000 (Thermo Scientific, Carlsbad, CA, USA) were added to each well. After 6 h of incubation, the medium was changed to complete medium. Twenty-four hours after transfection, the cells were subjected to cell migration, invasion, and drug sensitivity assays, and were harvested for RNA extraction and subsequent RT-qPCR.

### 2.11. Statistical Analysis

The results are displayed as mean ± standard deviation (SD). The significance of comparisons between groups was evaluated using the two-tailed Student’s *t*-test, with a significance level set at *p* < 0.05. Each experiment was performed with at least three replicates.

## 3. Results

### 3.1. Drug-Resistant CCA Cell Lines Exhibit Aggressive Behaviors

We established two chemotherapeutic drug-resistant CCA cell lines, namely KKU-213FR and KKU-213GR, representing the KKU-213A cells resistant to 5-FU and gemcitabine, respectively. These drug-resistant CCA cell lines underwent drug sensitivity assays under various concentrations of 5-FU and gemcitabine. The results confirmed that KKU-213FR cells were resistant to 5-FU, and KKU-213GR cells were resistant to gemcitabine, respectively (Figure 1a,b).

The Boyden chamber migration assays demonstrated that both drug-resistant cell lines exhibited significantly higher migration capabilities than the parental cell line (Figure 1c). Similarly, the invasion assay revealed that both KKU-213FR and KKU-213GR had higher invasion rates compared to their parental counterparts (Figure 1d).

### 3.2. Comparative Proteomic Analysis of Drug-Resistant CCA Cell Lines

We employed liquid chromatography with tandem mass spectrometry (LC-MS/MS) to conduct a comparative proteomic analysis of the parental CCA cell line (KKU-213A) and the drug-resistant cell lines (KKU-213FR and KKU-213GR). The proteomics analysis workflow is depicted in Figure 2a. In total, 5444, 5894, and 5848 proteins were identified from KKU-213A, KKU-213FR, and KKU-213GR, respectively. The partial least squares-discriminant analysis (PLS-DA) illustrated a good separation of identified proteins among groups (Figure 2b). Proteins with expression differences exceeding 3-fold between drug-resistant cell lines and the parental line, with a false discovery rate (FDR) < 0.05, were classified as differentially expressed proteins (DEPs). In KKU-213A vs. KKU-213FR, we identified a total of 279 DEPs, including 137 upregulated and 142 downregulated proteins. In KKU-213A vs. KKU-213GR, we identified 127 DEPs, comprised of 112 upregulated and 15 downregulated proteins. To identify important genes and pathways in these drug-resistant cells, we searched for overlapping DEPs and found 81 DEPs shared by both KKU-213FR and KKU-213GR, with 80 proteins upregulated (Figure 2c) and only 1 protein downregulated (Figure 2d). The log2 expression values of these proteins in KKU-213A, KKU-213FR, and KKU-213GR were visualized using a heatmap (Figure 2e).

### 3.3. Gene Ontology Analysis

To gain an overall perspective on proteome changes following the development of drug resistance, we subjected the list of DEPs from proteomics data to Gene Ontology (GO) analysis based on the GO database and protein analysis Through Evolutionary Relationships (PANTHER) classification. This allowed us to categorize these proteins based on their molecular function, biological processes, and protein classes (Figure 3). As depicted in Figure 3a, the largest fraction of the upregulated proteins is associated with the molecular function of binding proteins. Other prominent groups of proteins that exhibited changes include those with catalytic activity, transcriptional regulation, molecular transduction, and transporter activity. Furthermore, the analysis of biological processes revealed that most proteins were involved in cellular processes, followed by metabolic processes, localization, and development processes. Smaller percentages of proteins fell into categories related to responses to stimuli, multicellular organismal processes, signaling, and locomotion (Figure 3b). Additionally, these upregulated proteins were mainly classified into categories such as metabolite interconversion enzymes, transcriptional regulators, protein-modifying enzymes, and cytoskeleton proteins (Figure 3c).

### 3.4. The mRNA Expression of Upregulated Proteins in CCA Patients’ Tissues

To investigate whether the 80 upregulated proteins shared in both KKU-213FR and KKU-213GR were also upregulated in CCA patients’ tissues compared to their corresponding normal tissue counterparts, we utilized the UALCAN web portal as an analysis tool based on TCGA data. The results reveal that 25 out of 80 genes (Figure 4a) were significantly upregulated in CCA patients’ tissues compared with adjacent normal tissues, as shown in the heatmap in Figure 4b. These include TRAP1, CUL3, ARIH1, RPN2, NOTCH2, CDH2, TTLL4, PSMD13, DHX9, WDR74, NDRG1, MET, COMMD3, TERF2, TCP1, DDX19B, DMTF1, EGLN1, LAMB1, SNRPG, TMEM115, ITGA3, GRINA, RCOR3, and TJAP1 (Figure 4b).

### 3.5. Identification of Protein–Protein Interaction Networks

The 25 upregulated proteins were further investigated for protein–protein interactions (PPI) using STRING analysis. The results are represented in Figure 4c. The PPI network contains 23 nodes with 14 edges (versus 6 expected edges), a clustering coefficient of 0.514, an enrichment *p*-value of 0.00611, and an average node degree of 1.22, revealing the largest protein network that includes MET, LAMB1, ITGA3, NOTCH2, CDH2, and NDRG1. Smaller networks were also identified, including CUL3, COMMD3, ARIH1, and EGLN1. Additionally, connecting proteins such as DDX19B, DHX9, and SNRPG, along with two more connecting proteins, TCP1 and TRAP1, were identified. Subsequent analyses focused on the largest protein network, designated as the ‘focused network’. The mRNA expression of six genes in the focused network was further analyzed in CCA patients’ samples through the GEPIA2 web portal and verified in drug-resistant CCA cell lines using RT-qPCR. The results, presented as box plots in Figure 4d, demonstrate that LAMB1, ITGA3, NOTCH2, and NDRG1 exhibit significantly higher mRNA expression in CCA tissues compared to normal tissues, while the difference was not significant for MET and CDH2. Furthermore, the mRNA levels of all six genes were confirmed to be upregulated in both KKU-213GR and KKU-213FR (Figure 4e).

### 3.6. Assessment of the Prognostic Value of Six Selected Genes

We evaluated the correlation between the overall survival and disease-free survival of CCA patients and the mRNA expression levels of MET, LAMB1, ITGA3, NOTCH2, CDH2, and NDRG1 using TCGA data through the GEPIA2 web portal. No significant correlations were observed, possibly due to the small sample size of CCA patients’ tissues (Appendix A). We further explored the prognostic value of these six proteins in a pooled dataset of GI tract cancer patients, including cholangiocarcinoma (CHOL), colon adenocarcinoma (COAD), esophageal carcinoma (ESCA), liver hepatocellular carcinoma (LIHC), pancreatic adenocarcinoma (PAAD), rectum adenocarcinoma (READ), and stomach adenocarcinoma (STAD). The results are presented in survival maps (Figure 5a,b) and Kaplan–Meier plots (Figure 5c,d). High expression levels of LAMB1, ITGA3, NOTCH2, CDH2, and NDRG1 are associated with poor overall survival of GI tract cancer patients (Figure 5c). Additionally, elevated expressions of ITGA3, NOTCH2, and CDH2 are indicative of a shorter disease-free survival time (Figure 5d).

### 3.7. Correlation among Six Genes in the Focused Network

To further explore whether the connected proteins found in STRING analysis exhibit correlation in patients’ samples, we investigated the correlation of gene expression between connected proteins in the focused network using the Pearson correlation coefficient through the GEPIA2 platform. Except for CDH2 vs. NDRG1, we identified significant positive correlations between 10 gene pairs in GI tract cancer patients’ data, including MET vs. LAMB1, MET vs. ITGA3, MET vs. NOTCH2, MET vs. NDRG1, LAMB1 vs. ITGA3, LAMB1 vs. NOTCH2, LAMB1 vs. NDRG1, ITGA3 vs. NOTCH2, ITGA3 vs. NDRG1, and NOTCH2 vs. NDRG1. On the other hand, four gene pairs, including MET vs. CDH2, LAMB1 vs. CDH2, ITGA3 vs. CDH2, and CDH2 vs. NDRG1, showed negative correlations (Figure 6). Additionally, when analyzing the CHOL dataset alone, we found positive correlations between five gene pairs in CCA patients’ data, including MET vs. LAMB1, MET vs. ITGA3, MET vs. NOTCH2, MET vs. CDH2, and NOTCH2 vs. CDH2 (Appendix A).

### 3.8. siRNA-Mediated Knockdown of Six Selected Genes Attenuates Cell Migration, Cell Invasion, and Reverses Drug-Resistant Phenotypes

We further confirmed the significance of the six selected genes using a loss-of-function study with siRNA. Gene knockdown efficiency was confirmed with RT-qPCR, and the results showed that siMET, siLAMB1, siITGA3, siNOTCH2, siCDH2, and siNDRG1 could significantly and effectively suppress the mRNA expression of the respective genes (Figure 7a). The drug-resistant CCA cells (KKU-213FR and KKU-213GR) transfected with all six gene-specific siRNAs showed reduced cell migration (Figure 7b) and cell invasion (Figure 7c) capability when compared to those cells transfected with siNC. Moreover, knockdown of these six genes suppressed the drug-resistant phenotype of these cells (Figure 7d).

## 4. Discussion

Despite recent advancements in surgical techniques, chemotherapy, and the development of targeted therapies at the molecular level, the primary cause of cancer-related deaths is still the progressive growth of metastases that show resistance to existing treatments [26]. Although chemotherapy is a commonly employed approach for managing advanced CCA patients, even with combinations of drugs, the median survival rate remains below one year [27]. This challenge arises from the intricate mechanisms of chemoresistance, which typically aid cancer cells in evading the lethal effects of anticancer drugs. Thus, gaining a deeper comprehension of the molecular foundations of drug resistance is crucial for unraveling the complex mechanisms and indicators associated with drug-resistant characteristics.

Previous research has successfully developed cell lines that are resistant to commonly used chemotherapeutic drugs 5-FU and gemcitabine in various cancer types, and these cell lines have become invaluable tools for elucidating drug resistance mechanisms and identifying potential therapeutic targets in cancer research. For instance, 5-FU resistance has been achieved in hepatocellular carcinoma cells (HLF-cell line) [28], colorectal cancer cells (HCT-8 and H630) [29,30], breast cancer cells (T47D) [30], triple-negative breast cancer cells (MDA-MB-231) [31], and gastric cancer cells (SNU638) [32]. In pancreatic cancer, gemcitabine-resistant cell lines have been established, such as SW1990 [33], PaCa-2 [34], PANC [35], and in pancreatic ductal carcinoma cell lines [36]. Additionally, gemcitabine-resistant cell lines have also been developed in CCA, specifically KKU-M139 and KKU-M214 cell lines [18]. In this research, we successfully established 5-FU- and gemcitabine-resistant CCA cell lines KKU-213FR and KKU-213GR (Figure 1a,b).

The intricate process of drug resistance in cancer cells linked to metastasis-related differentiation signaling involves dynamic changes in both cancer and stromal cells [37]. Metastasis encompasses a decrease in tumor cell adhesion receptor expression, promoting heightened cell motility, while invasion entails the disassembly of the extracellular matrix (ECM) [38]. Patients with chemotherapeutic resistance in CCA often undergo advanced cancer recurrence with metastasis [39]. Subsequently, we explored the metastatic potential of 5-FU- and gemcitabine-resistant CCA cell lines. Further assessment through the Boyden chamber assay demonstrated a higher number of migrated and invaded cells, affirming the heightened metastatic potential of the established drug-resistant cell line (Figure 1c,d).

To investigate alterations in cellular mechanisms related to drug resistance, we identified changes in the proteome profiles. A total of 80 proteins were upregulated in both KKU-213FR and KKU-213GR when compared to the parental cell line. These proteins underwent classification using GO analysis, categorizing them based on their molecular function, biological process, and protein class. The upregulated proteins were primarily classified under the molecular function category of ‘binding’ (GO: 0005488). Previous studies have highlighted the significant roles played by various binding proteins in cancer progression and the development of drug resistance. These proteins function in diverse ways such as transmitting signals through signaling molecules [40,41], altering the number or mutation of receptors [42,43,44], and releasing cytokines [45,46].

Drug resistance mechanisms involve alterations in various cellular processes such as increased cell replication [47,48], transcriptional activation of drug-resistant genes or oncogenes [49,50], and DNA repair in response to exposure to chemotherapeutic drugs to prevent cell death [51,52,53].

The second largest group of proteins was categorized under biological regulation and metabolic processes. In response to chemotherapeutic agents, cancer cells that possess the ability to evade cell death caused by the drugs can alter their biological processes to sustain growth or proliferation [54,55], migrate to other parts of the body by reducing cell adhesion [56,57], and secrete enzymes or signals to modulate the extracellular matrix [58,59].

When categorized via protein class, the majority of upregulated proteins were classified as metabolite interconversion enzymes. Metabolic reprogramming of cancer cells can hinder immune responses and create obstacles to cancer treatment. Cancer cells must balance increasing metabolic demands associated with uncontrolled cell division and metastasis, which often render them resistant to current therapies. Several studies have discovered abnormalities in metabolism in chemotherapy-resistant cancers [60,61,62].

We further compared upregulated proteins from the proteomic data to their mRNA expression in patients’ tissues from the TCGA database through the UALCAN platform [24]. Among the 80 upregulated proteins identified in drug-resistant cell lines based on proteomics analysis, only 25 were found to be upregulated in cholangiocarcinoma (CCA) tissues compared to normal tissues when analyzed using TCGA data through the UALCAN web portal. This discrepancy may be attributed to the absence of comparative gene expression data between chemosensitive and chemoresistant CCA patient samples within the CHOL dataset. The 25 overlapping genes may play a crucial role in both drug resistance and carcinogenesis in CCA, whereas the remaining 56 non-overlapping genes may solely contribute to the development of drug resistance. Future research involving the analysis of these 80 upregulated genes in drug-sensitive and drug-resistant clinical samples will further elucidate their roles in drug-resistant CCA. When subjected to STRING analysis, we identified a highly interconnected network comprising six proteins: MET, LAMB1, ITGA3, NOTCH2, CDH2, and NDRG1. It is notable that a medium confidence level (0.4) was used as a parameter in the STRING analysis. If high confidence (0.7) was used, the protein network would remain largely similar, except for the exclusion of NOTCH2 and NDRG1. To avoid excluding significant genes, the medium confidence value was utilized. Nearly all of the proteins in this network, except for MET, were associated with the poor prognosis in GI tract cancer patients (Figure 5). Although statistical significance was not found when analyzing CCA data alone, it was likely due to the small sample size. Furthermore, the mRNA expression in patients’ samples of genes in the focused network was positively correlated with each other (Figure 6). Interestingly, the silencing of these six genes reduces cell motility rates and attenuates the chemotherapeutic-resistant phenotype of both KKU-213FR and KKU-213GR (Figure 7).

Several past investigations have demonstrated the oncogenic functions of these proteins in different cancer types. The mesenchymal epithelial transition factor (MET), alternatively termed hepatocyte growth factor receptor (HGFR), is a receptor tyrosine kinase. When it interacts with its ligand, hepatocyte growth factor triggers various cellular signaling pathways related to proliferation, motility, migration, and invasion [63,64].

Laminin beta-1 (LAMB1), a type of extracellular matrix (ECM) glycoprotein, is present in most tissues and is responsible for initiating cell assembly. This assembly is crucial for the invasion and spread of cancer cells [65]. Studies have shown that LAMB1 is significantly expressed in multiple invasive cancers [66] and plays a significant role in cancer progression, as observed in gastric cancer [67,68], and lung adenocarcinoma [66].

Integrin alpha-3 (ITGA3) is a cell membrane-bound integrin protein receptor that plays a role within the ECM. Its function involves serving as a cell surface adhesion molecule and engaging with various ECM-receptor proteins such as integrin beta-1 (ITGB1), members of the laminin family, and fibronectin 1 [69].

Neurogenic locus notch homolog protein 2 (NOTCH2) is a product of the *NOTCH2* gene. It plays dual roles in cell interactions and serves as a highly conserved signal transduction system once its ligand binds. Overexpression of NOTCH2 has been identified in multiple cancer types including breast cancer [70], lung squamous cell carcinoma [71], leukemia [72], ovarian cancer [73], and colorectal cancer [74]. The activation of NOTCH2 is linked to various mechanisms in tumor development such as regulating the properties of tumor-initiating cells, controlling signaling pathways like MYC [75] or P53 [76,77], promoting angiogenesis [78], regulating tumor invasion [79,80], and managing the cell cycle [81,82].

CDH2, also known as neuronal cadherin (N-cadherin), is a marker of epithelial-to-mesenchymal transition (EMT). CDH2 is referred to as a ‘mesenchymal cadherin’, which replaces epithelial cadherin (E-cadherin) during the epithelial-to-mesenchymal transition known as the cadherin ‘switch’ [83,84]. CDH2’s functions are associated with fibroblast growth factor receptors (FGFRs). This interaction leads to continuous FGFR signaling and the progression of carcinoma [85]. Additionally, CDH2 enhances the activity of nuclear β-catenin-mediated drug resistance in myeloma [86]. The roles of CDH2 have been demonstrated in several cancers such as increasing invasiveness in melanoma cancer [87] and esophageal squamous cell carcinomas [88].

N-myc downregulated gene-1 (NDRG1) is known to have diverse roles, functioning both as a suppressor of metastasis and as an indicator of poor prognosis, while also contributing to disease progression across various cancer types [89]. Although NDRG1 is primarily recognized for its anti-oncogenic and anti-metastatic functions [90,91,92], studies have revealed its pro-oncogenic role in certain cancers like gastric cancer and hepatocellular carcinoma [93]. In clinical samples of esophageal cancer, elevated levels of NDRG1 were associated with the malignant advancement of the disease [94]. This study proposed that NDRG1 influences the Wnt signaling pathway and the accumulation of β-catenin through the mediation of Wnt-associated genes, thereby promoting metastasis [83]. Indeed, reducing NDRG1 levels in gastric cancer cells led to increased E-cadherin expression and decreased vimentin expression, indicating a link between high NDRG1 levels and the metastatic potential of gastric cancer cells [95]. Overall, the role of NDRG1 in either promoting or suppressing tumor progression is highly dependent on the specific type of tumor cell and its degree of differentiation.

Taken together, MET, LAMB1, ITGA3, NOTCH2, CDH2, and NDRG1 were upregulated in drug-resistant CCA cell lines and in CCA patients’ tissues. These genes not only contribute to acquired drug resistance but also play a crucial role in the progression of CCA cells. They share a common role in activating tumor metastasis through EMT pathways. Figure 8 provides a schematic picture illustrating the predicted functional role of these proteins in the development of drug resistance and progression of CCA.

## 5. Conclusions

Despite advancements in cancer therapies, metastasis-driven drug resistance poses a formidable challenge leading to high mortality rates. Our study successfully established 5-FU- and gemcitabine-resistant CCA cell lines KKU-213FR and KKU-213GR, which exhibited increased cell motility and invasion. Through the integration of proteomics analysis and bioinformatics, we identified 25 genes that were upregulated in drug-resistant cell lines as well as in patients’ tissues. Further investigation revealed a network of six highly interconnected genes (MET, LAMB1, ITGA3, NOTCH2, CDH2, and NDRG1) whose expressions are correlated with each other. Furthermore, the mRNA expression of these genes is associated with a poor prognosis in GI tract cancer patients, emphasizing their potential as prognostic markers and possibly therapeutic targets.

## Figures and Tables

**Figure 1 biomolecules-14-00969-f001:**
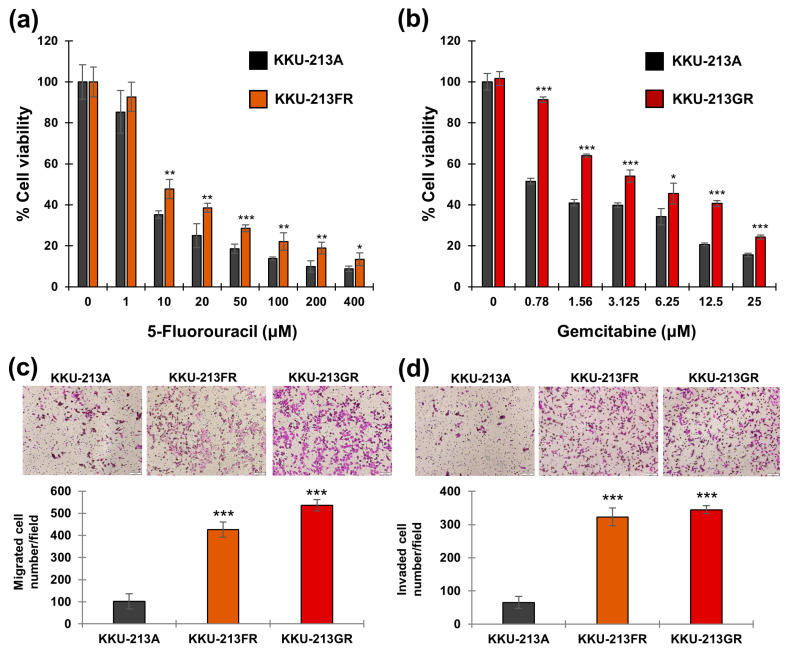
Phenotypic study of drug-resistant CCA cell lines. Drug sensitivity assay of KKU-213FR (**a**) and KKU-213GR (**b**). (**c**) Cell migration assay. (**d**) Cell invasion assay: magnification 100×; scale bar, 100 μm. The data are presented as mean ± SD from three replicates. Significant differences between treated versus untreated controls or parental versus drug-resistant cells are indicated by * *p* < 0.05, ** *p* < 0.01, and *** *p* < 0.001.

**Figure 2 biomolecules-14-00969-f002:**
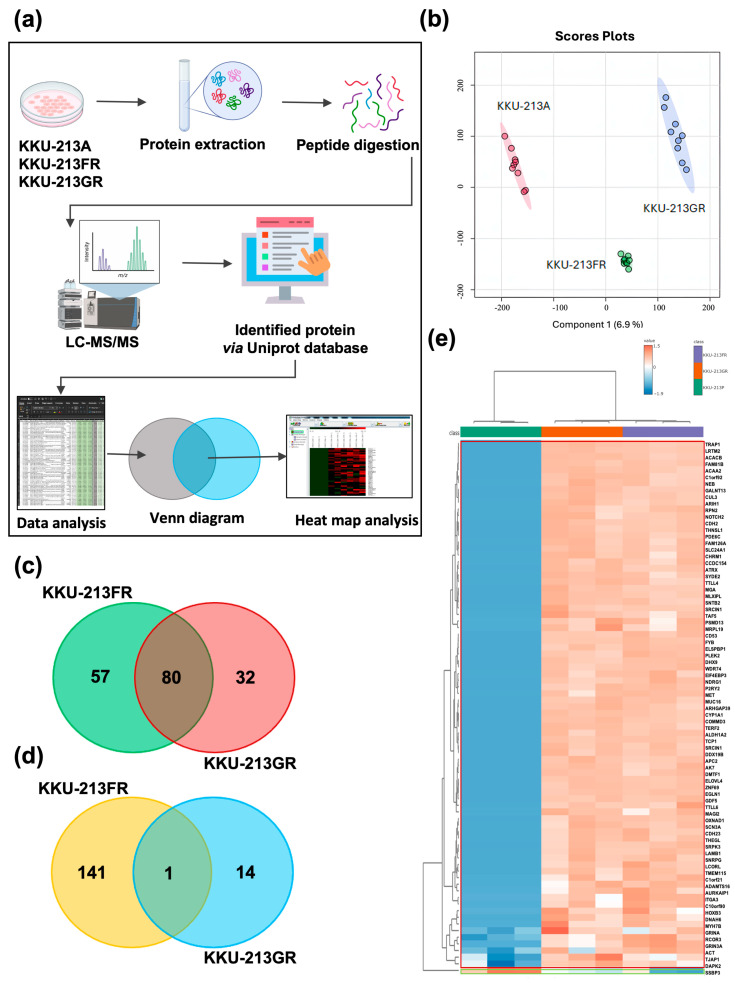
Proteomics analysis of KKU-213A, KKU-213FR, and KKU-213GR cell lines. (**a**) Proteomics analysis workflow. (**b**) Partial least squares-discriminant analysis (PLS-DA) of all identified proteins. Venn diagram of upregulated proteins (**c**) and down-regulated proteins (**d**) in KKU-213FR and KKU-213GR compared to KKU-213A. (**e**) Heatmap with group averages of upregulated (red frame) and down-regulated (green frame) proteins shared among KKU-213A, KKU-213FR, and KKU-213GR.

**Figure 3 biomolecules-14-00969-f003:**
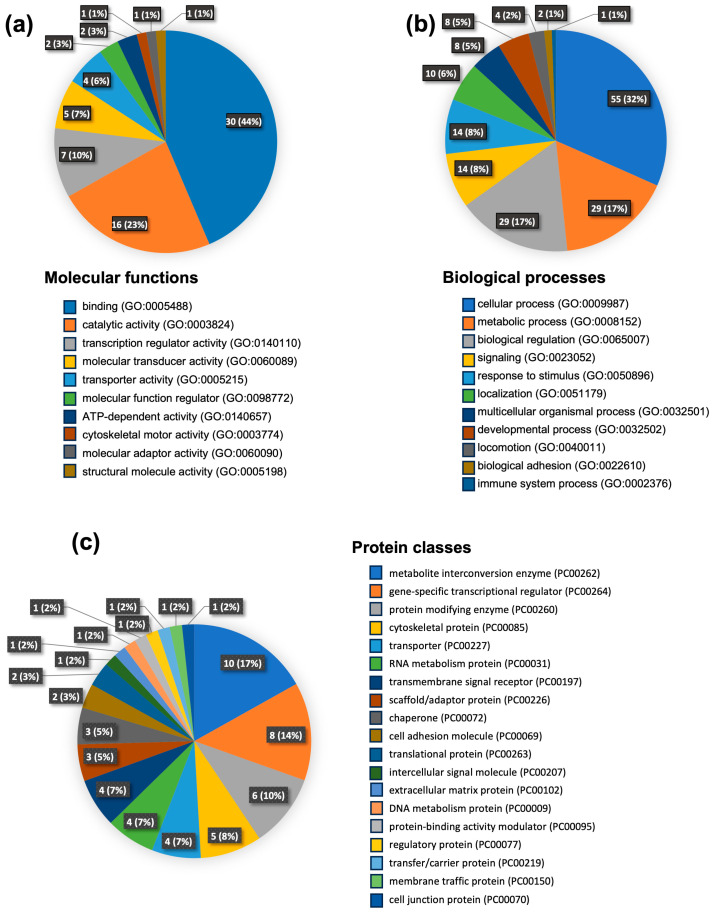
Gene Ontology analysis of upregulated proteins shared in KKU-213FR and KKU-213GR. The analysis represents the proteins hits and their respective percentages. Analysis based on molecular functions (**a**), biological processes (**b**), and protein classes (**c**).

**Figure 4 biomolecules-14-00969-f004:**
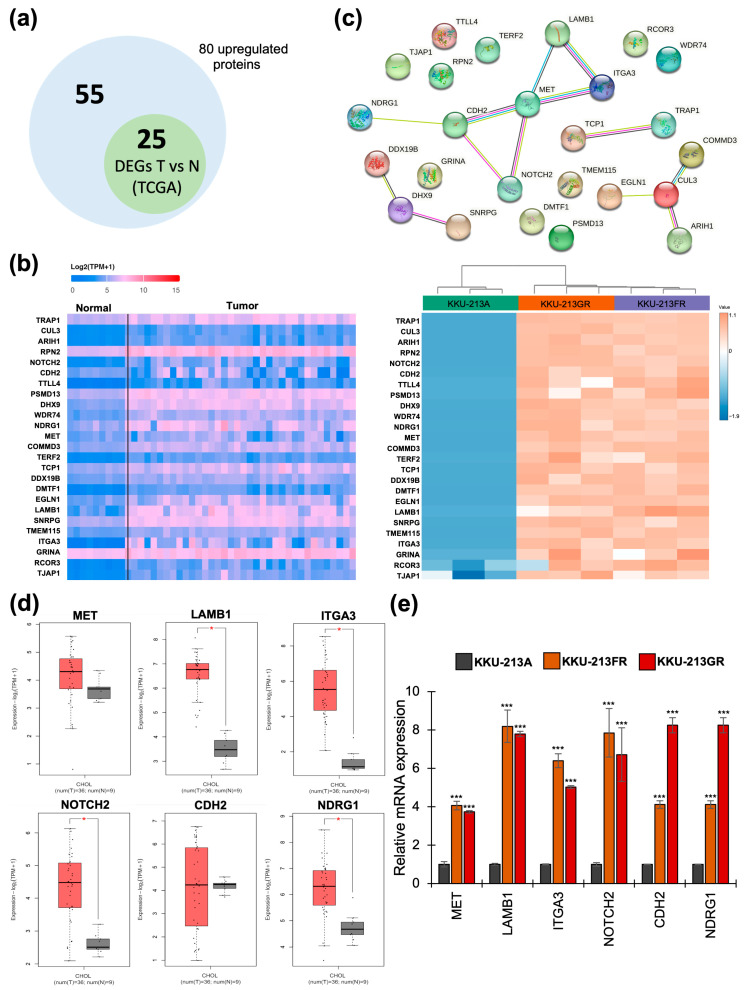
Identification of significant genes in CCA. (**a**) The Venn diagram represents 80 upregulated proteins from the proteomic analysis and 25 upregulated genes in CCA tissue compared to normal tissue. (**b**) Heatmap analysis displaying the expression of 25 overlapping genes that were significantly upregulated in CCA patients’ tissues based on TCGA data (left) and proteomics data (right). (**c**) PPI analysis of the 25 upregulated genes via STRING. (**d**) Box plot showing the mRNA expression of six selected genes (a focused network) in CCA patients’ tissues (red box) vs. normal tissues (grey box) based on TCGA data. Significant differences between groups are indicated by * *p* < 0.05. (**e**) The mRNA expression of six selected genes verified by RT-qPCR in KKU-213A, KKU-213FR, and KKU-213GR. Significant differences between the KKU-213A and KKU-213FR groups, as well as between the KKU-213A and KKU-213GR groups, are indicated by *** *p* < 0.001.

**Figure 5 biomolecules-14-00969-f005:**
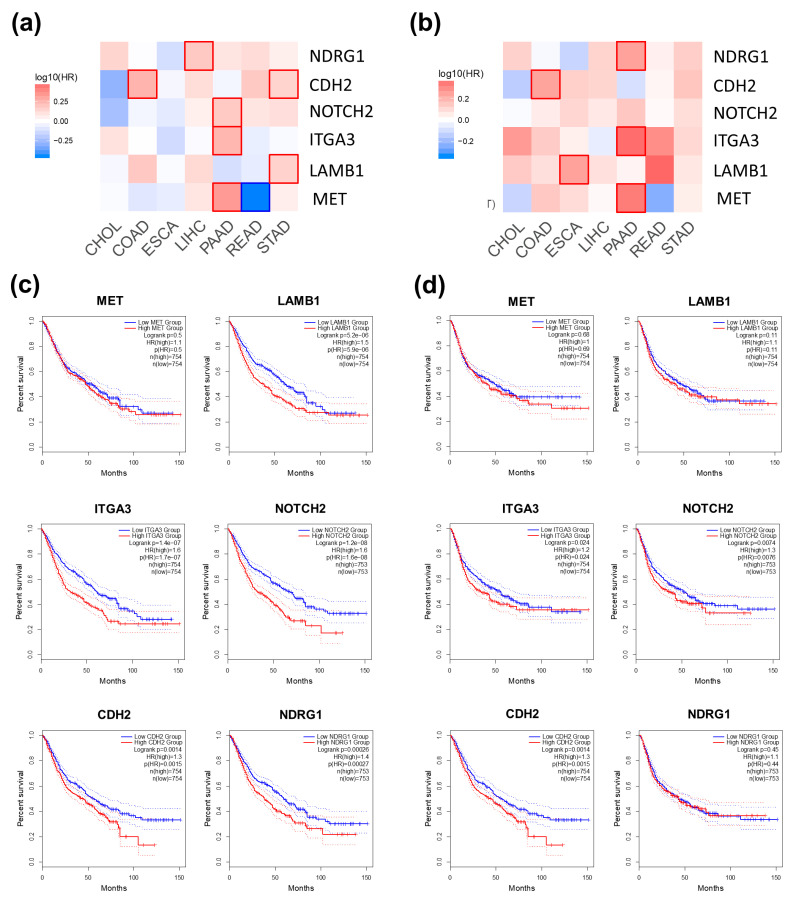
Survival analyses of six selected genes in GI tract cancer patients. The overall survival map (**a**) and disease-free survival map (**b**) are displayed. Red shading indicates poor prognosis, and blue shading indicates good prognostic potential for each gene. The box with a prominent border represents statistical significance. Kaplan–Meier plots are used to represent overall survival (**c**) and disease-free survival (**d**).

**Figure 6 biomolecules-14-00969-f006:**
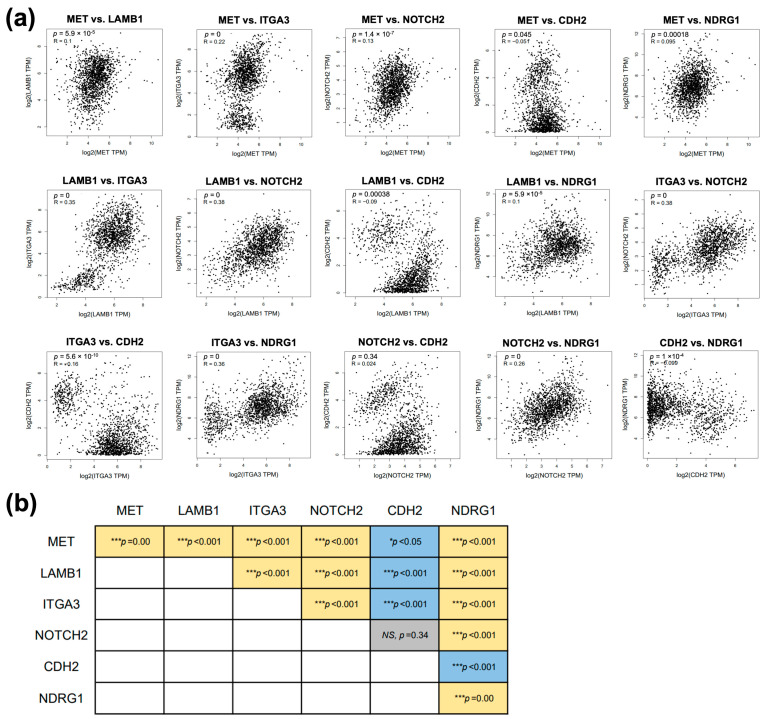
The correlation analysis of six focused genes in GI tract cancers includes. (**a**) Scatter plots depicting the expression correlation between gene pairs in GI tract cancers. (**b**) Summary of the significant correlations among gene pairs in GI tract cancers. Yellow color indicates a positive correlation, blue color indicates a negative correlation, and grey color indicates no significant correlation between genes. Significant correlations between gene pairs are indicated by *** *p* < 0.001, while *NS* denotes non-significance.

**Figure 7 biomolecules-14-00969-f007:**
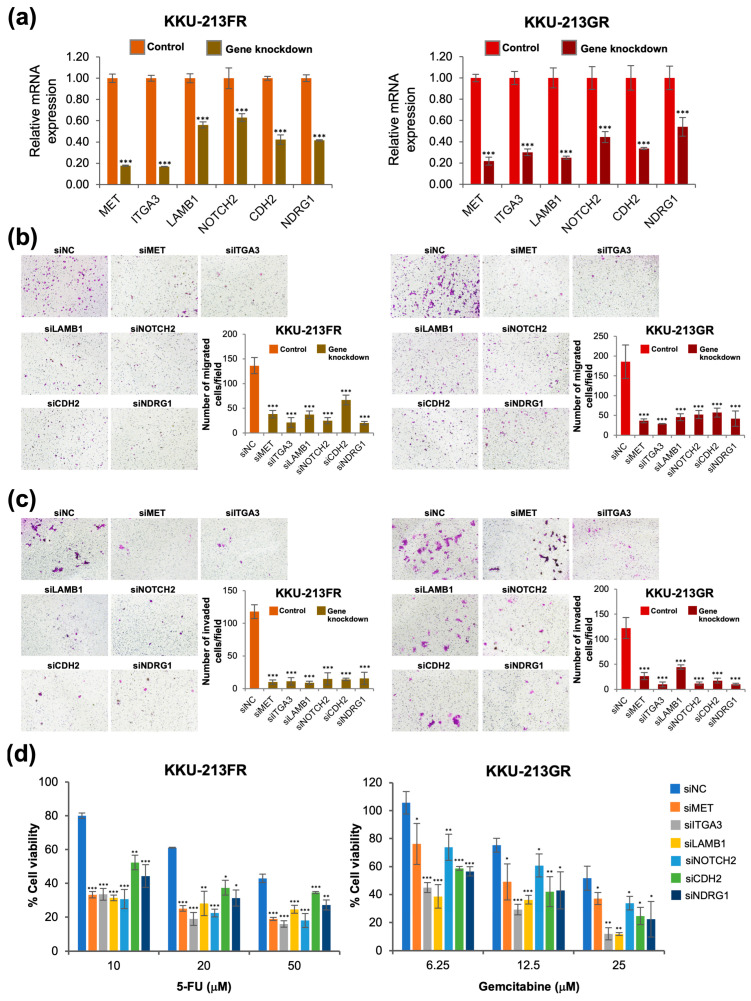
Phenotypic study after knockdown of six selected genes in KKU-213FR and KKU-213GR cells. (**a**) mRNA expression after siRNA transfection. (**b**) Cell migration assay. (**c**) Cell invasion assay. (**d**) Drug sensitivity assay. Data are presented as mean ± SD from three replicates. Significant differences between siRNA-transfected cells and siNC-transfected controls are indicated by * *p* < 0.05, ** *p* < 0.01, and *** *p* < 0.001, magnification 100×; scale bar, 100 μm.

**Figure 8 biomolecules-14-00969-f008:**
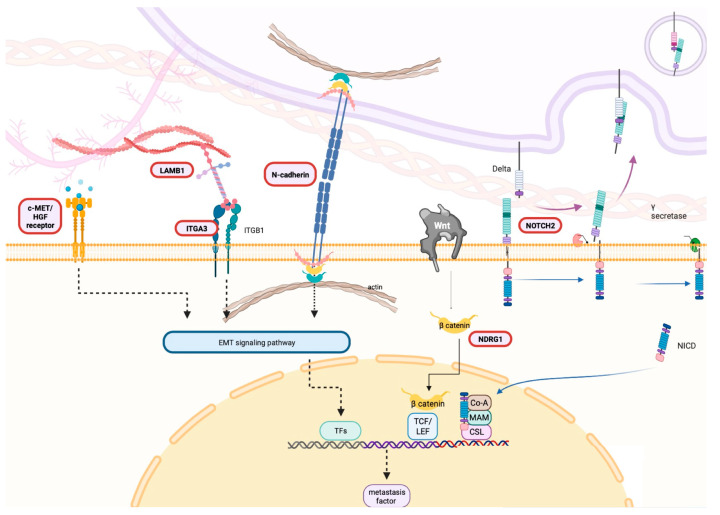
Hypothetical role of six selected genes in CCA progression. A schematic illustration delineating the putative role of the selected genes, elucidating their hypothesized involvement in driving drug resistance and promoting the progression of CCA cells. This figure was constructed using BioRender (2023).

**Table 1 biomolecules-14-00969-t001:** Lists of gene-specific primers for RT-qPCR.

Gene	Sequence (5′ to 3′)	
*ACTB*	For:	GGATTCCTATGTGGGCGACG
	Rev:	TTGTAGAAGGTGTGGTGCCAG
*MET*	For:	CGCACAAAGCAAGCCAGATT
	Rev:	AGTGCTCATGATTGGGTCCG
*LAMB1*	For:	GGCAATCTGAAAATGGTGTGGA
	Rev:	ACGAGGCCTCACAGTCATAG
*ITGA3*	For:	GGGACAGTGATGGGTGAGTC
	Rev:	GTAGGGCCACTCCAGACCTA
*NOTCH2*	For:	AGGTGTCAGAATGGAGGGGT
	Rev:	GCCGTTGACACATACACAGC
*CDH2*	For:	TGCAAGACTGGATTTCCTGAAGA
	Rev:	AGCTTCTCACGGCATACACC
*NDRG1*	For:	ATTGGCATGGGAACAGGAGC
	Rev:	CATCCTGAGATCTTGGAGGCG

**Table 2 biomolecules-14-00969-t002:** List of siRNAs for gene knockdown.

siRNA	Target Gene	siRNA Sequence (5′ to 3′)
siMET	MET	GGACCGGUUCAUCAACUUCTT
siLAMB1	LAMB1	AAUGUAACUGCAAUGAACATT
siITGA3	ITGA3	GGAAAGGAAACAGCUACAUGATT
siNOTCH2	NOTCH2	GAAUUGUCAGACAGUAUUGTT
siCDH2	CDH2	UGACAACAGACCUGAGUUCTT
siNDRG1	NDRG1	GACCACUCUCCUCAAGAUGTT
siNC	-	UUCUCCGAACGUGUCACGUTT

## Data Availability

All data generated or analyzed during this study are included in this published article and its Appendix A. The MS/MS raw data and analysis have been deposited in the ProteomeXchange Consortium via the jPOST partner repository with the data set identifier JPST002506 and PXD049309 (https://proteomecentral.proteomexchange.org/cgi/GetDataset?ID=PXD049309, accessed on 29 July 2024).

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
