# Peer review of "Proteomics and Bioinformatics Identify Drug-Resistant-Related Genes with Prognostic Potential in Cholangiocarcinoma"

_biomolecules, 2024, doi:10.3390/biom14080969_

Round 1
Reviewer 1 Report
Comments and Suggestions for Authors
The manuscript is potentially interesting and presents a description of differences between drug-resistance CCA cell lines. The authors declarate the potential molecular mechanisms underlying drug resistance and tumor progression in CCA.
However, there are still areas that require clarification and improvement. One such area is the detailed presentation (in figures and tables) of the results from analyses.
First, the quality control for proteomic profiling.
Second, there isn't information about replication number and their comparison. According to link in ProteomeXchange, there are 3 replications, without their analysis in the manuscript.
Also, the authors should more detailed description parameters of PPI analysis by STRING (score, for example).
In the introduction, it is worth immediately indicating the nature of the KKU-213A cells.
When analyzing the TCGA data, it is not entirely clear that 25 genes are up-regulated according to cell line comparisons.
Reviewer 2 Report
Comments and Suggestions for Authors
In the article ‘Proteomics and Bioinformatics Identify Drug-Resistant-Related 2 Genes with Prognostic Potential in Cholangiocarcinoma' Kerdkumthong et al used a cholangiocarcinoma cell line, made it resistant to 5 fluro-uracil and gemcitabine and performed proteomics analysis to detect regulated proteins. I have a few concerns about this study:
1) I couldn’t download any supplementary files and saw no mention of them in the main text. Authors should upload supplementary table for all identified proteins, all parameters used for the search and from DeCyderMS, filteres regulated proteins. Also needs to be a supplementary file of the GO/Panther enrichment results listing p value, enrichment factor, number of proteins in each annotation etc.
2) Authors should be uploading raw files to Proteome Xchange or similar for raw files to be easily analysable by others. This is the norm these days for any large scale data.
3) Authors need to detail what label free analysis approach was used to quantitate protein abundance. How was the correlation of results between the replicates? Can they provide a scatter plot and Pearson’s correlation coefficients in a supplementary figure?
4) Figure 4 and the analysis in section 3.4 needs to be improved. Figure 4 a,b,c is TCGA data and not authors data. Also the authors identified 169 upregulated and 156 downregulated proteins (I calculated this from the venn diagrams) and then over lapped them with the CCA dataset from the TCGA and identifies how many overlapping proteins following similar trends in their proteomics data and the TCGA data is not clear. Are the 25 proteins listed in the heatmap in 4a also upregulated in their proteomics datatset? Why do they then look at mRNA levels of these 6 genes and not the protein level expression from the proteomics? It makes no sense to have done proteomics analyses if the data correlation of the TGCA data is not shown to their proteomics data. Please show correlation of these genes with proteomics or perhaps add a corresponding protein heatmap showing similar or dissimilar trends. Also I am wondering if the authors could widen the list of genes found regulated in CCA from more recent large scale datasets that have been published. (https://www.mcponline.org/article/S1535-9476(23)00115-9/fulltext, https://www.sciencedirect.com/science/article/pii/S1535610821006590?via%3Dihub, https://journals.lww.com/hep/fulltext/2023/02000/proteogenomic_characterization_of.11.aspx)
5) STRING interaction network would be better if performed with their regulated list of proteins from the proteomics data. Also the resolution of the STRING screenshot is poor. Consider using cytoscape to render these interactions.
6) Can the authors perform knockdown of one these 6 proteins in the resistant cell line and show that it reduces invasion, migration to show that they are relevant? This would greatly strengthen all their data.
Minor issues:
1) Line 135, mention fasta file date/version used for analysis.
2) Under RT-PCR methods what house-keeping gene was used and what was the primer sequence?
3) In lines 223-225, were the samples run in biological or technical triplicate?
4) In figure 2 in the heatmap, please mark the proteins that are up and down regulated.
5) In figure 3 in the pie chart rather than having the GO IDs in brackets, it would be rather better to have number pf proteins in that annotation. Readers will benefit from knowing how many of these listed regulate proteins are in each annotation.
6) Figure 5c and 5d and 6a have poor resolution. Difficult to read text in the figures.
Comments on the Quality of English LanguageMinor editing of English language required
Reviewer 3 Report
Comments and Suggestions for Authors
The authors used proteomic and bioinformatic tools to identify drug-resistant-related 2 proteins in cholangiocarcinoma. The overall experimental design and execution are valid, but lack several important details for the accurate evaluation of the manuscript.
1. Line 133, the authors stated analysing the same sample three times using LC-MS. Does it mean there is no biological replicates at all, but only the three technical replicates for the study? If so, this is very odd and hard to justify.
2. Line 135, please indicate the date/version of the Uniprot protein database.
3. With a relatively old ion-trap MS, identifying over 5000 proteins with 30 minutes gradient seems to be very realistic. Please provide more details on the valication of the protein identification results (filters, algorithm, FDR etc).
4. Line 143, please specify the DEPs were seleted based on what creteria, spectral counting? precursor ion area?
5. The heatmap is somewhat difficult to understand as to how to differentiate between upregulated and downregulated proteins? It looks to me all proteins are upregulated based on the color.
6. Please provide vocalno plots to give an distinct overview of the data.
Round 2
Reviewer 1 Report
Comments and Suggestions for Authors
The authors have done a good job of revising the article. I have only one comment regarding the parameters used for PPI abalysis. For STRING database the unspoken confidence score is 0.700, not 0.400. In view of this weak parameter, the authors need to justify this choice.
Author Response
Thank you for your insightful feedback. We initially utilized the default setting of STRING for our analysis, which employed a confidence score of 0.400. Upon your recommendation, we reanalyzed the data using a confidence score of 0.700 and found that the protein network remained largely similar, except that NOTCH2 and NDRG1 were not included. According to TCGA data, NOTCH2 and NDRG1 are positively correlated with other genes in the network. Our experimental results also highlight the significance of these genes in drug resistance and the progression of CCA. Therefore, we believe using a medium confidence score of 0.400 is justified to avoid excluding these important genes. We have also discussed this point in the discussion section, page 17, lines 497-501.
“It is notable that a medium confidence level (0.4) was used as a parameter in the STRING analysis. If high confidence (0.7) were used, the protein network would remain largely similar, except for the exclusion of NOTCH2 and NDRG1. To avoid excluding significant genes, the medium confidence value was utilized.”
Reviewer 2 Report
Comments and Suggestions for Authors
The authors have greatly improved the manuscript. My concerns are:
1) While the supplementary files are downloadable what is missing is as follows. Authors should upload supplementary table for all identified proteins, all parameters used for the search and from DeCyderMS, filtered regulated proteins. Also needs to be a supplementary file of the GO/Panther enrichment results listing p value, enrichment factor, number of proteins in each annotation etc.
2) Regarding the analysis in figure 4, The overlap of the data from proteomics and the public data is 25 of 80 regulated proteins they identified. They should depict in figure 4b a corresponding heatmap of the proteome level changes and also provide a supplementary table with values for both heatmaps. This would be an additional supplementary table to the ones I mentioned above. In the discussion the authors should comment on why only 25 overlapping genes were found regulated.
Comments on the Quality of English LanguageEnglish language seems overall fine.
Author Response
We thank the reviewer for thoroughly reviewing our revised manuscript and for the valuable suggestions. We have now uploaded the following supplementary materials or incorporated them into the manuscript as requested:
1. A table listing all identified proteins (File name: RawData_all_proteins).
2. All parameters used for quantitation by DeCyderMS have been incorporated into the text (page 4, lines 157-169).
“Briefly, the raw LC-MS/MS data files were imported into the program, which transformed the information into a virtual peptide image. These images were then processed within the Pepdetect module of DecyderMS. The program initially identifies individual peptides by placing a box around them. Subsequently, ion counts for each peptide are integrated, and the Log2 of this number is reported. The process is repeated for each replicate sample. The resulting data are then uploaded to the Pepmatch module, where peptides within different virtual images are matched based on m/z and retention time. Notably, because the data were acquired in an ion trap mass spectrometer, tandem MS ensures accurate peptide matching (i.e., MS/MS spectra consistency). Within the Pepmatch module, each virtual image is assigned to a specific group (KKU-213A, KKU-213FR, and KKU-213GR). Group-to-group comparisons are performed. Integrated peptide counts serve as the basis for statistical analyses, assessing the probability that observed changes are not due to chance. False discovery rates are determined from the data.”
- A list of filtered regulated proteins includes 279 DEPs in KKU-213FR (137 up- and 142 downregulated proteins), 127 DEPs in KKU-213GR (112 up- and 15 downregulated proteins), and DEPs shared in both KKU-213FR and KKU-213GR, comprising 80 up- and 1 down-regulated proteins (File name: RawData_DEPs).
For clarity, we have added this information to the results section on page 7, lines 287-291.
“In KKU-213A vs. KKU-213FR, we identified a total of 279 DEPs, including 137 upregulated and 142 downregulated proteins. In KKU-213A vs. KKU-213GR, we identified 127 DEPs, comprised of 112 upregulated and 15 downregulated proteins. To identify important genes and pathways in these drug-resistant cells, we searched for overlapping DEPs and found 81 DEPs shared by both KKU-213FR and KKU-213GR, with 80 proteins upregulated (Figure 2c) and only 1 protein downregulated (Figure 2d). The log2 expression values of these proteins in KKU-213A, KKU-213FR, and KKU-213GR were visualized using a heatmap (Figure 2e).”
- We have provided a file detailing the GO/Panther results, including the number of proteins in each annotation (File name: RawData_GO).
However, p-values and enrichment factors were not available in this analysis. According to the PANTHER web portal, various types of analyses are available, such as the Statistical Enrichment Test, Statistical Overrepresentation Test, and Functional Classification. The Statistical Overrepresentation Test involves comparing your dataset against a reference gene list, providing detailed results including the number of proteins, fold enrichment, and p-values, which indicate the statistical significance of the overrepresentation of certain functional categories. Our results are based on the Functional Classification analysis, which focuses on the number of protein hits in gene ontology categories. This approach differs from enrichment analysis as it does not involve a reference gene list for comparison. Instead, it categorizes proteins based on their functions and reports the number of hits in each category. Consequently, our analysis provides the number of protein hits and calculates the percentage of total protein hits in each classification. This means our results do not include p-values or fold enrichment factors, as these are specific to enrichment analyses. We hope this explanation clarifies the differences in our analytical approach and the nature of the data provided.
We have depicted a corresponding heatmap of the proteome level changes in Figure 4b. Additionally, we have provided a supplementary table with the values for both heatmaps (File name: RawData_25_overlap_genes). Furthermore, we have added information to the discussion section explaining why only 25 out of the 80 upregulated proteins were found to overlap with the public data. This discussion can be found on page 17, lines 486-495.
“Among the 80 upregulated proteins identified in drug-resistant cell lines based on proteomics analysis, only 25 were found to be upregulated in cholangiocarcinoma (CCA) tissues compared to normal tissues when analyzed using TCGA data through the UALCAN web portal. This discrepancy may be attributed to the absence of comparative gene expression data between chemosensitive and chemoresistant CCA patient samples within the CHOL dataset. The 25 overlapping genes may play a crucial role in both drug resistance and carcinogenesis in CCA, whereas the remaining 56 non-overlapping genes may solely contribute to the development of drug resistance. Future research involving the analysis of these 80 upregulated genes in drug-sensitive and drug-resistant clinical samples will further elucidate their roles in drug-resistant CCA.”
Reviewer 3 Report
Comments and Suggestions for Authors
The authors provided satisfactory feedback/revisions to most of my comments. Yet, I still have one major concern about the unusually high number of identified proteins, using a low-resolution ion-trap MS, with a data acquisition rate of merely 2Hz (2 spectral per second) and a short analysis time of 30 minutes. Moreover, no prior fractionation of the complicated peptides samples was performed (2D-LC) to compensate for the low scan rate of the MS equipment. With such an equipment, a 30-minute gradient can only generate roughly 3600 spectral (MS1 and MS2 combined). The author did not specify if the MS data acquisition is a data-dependent type and how many MS2 spectral were following a MS1 spectral. Nevertheless, with only 3600 spectral, it is not likely at all to generate over 5000 (close to 6000) unique protein identification. One possible explanation could be that these protein IDs are not unique, or not grouped properly. Please see the Paris guidelines for details (https://www.mcponline.org/guidelines). Please upload or share the protein identification result for each sample/replicate with as many details as possible in Excel format for us to have a better idea on the number of reported proteins.
My 2nd concern is for the volcano plots. Proteins are confined to several small areas of the plots compared to traditional volcano plots. Do the authors have an explanation for this? Please share the data source (excel format) for the volcano plots.
Based on the volcano plots, there should be more than 80 DEPs. Please explain.
Author Response
We thank the reviewer for the thoughtful review of our revised manuscript. We have carefully addressed your concerns as follows:
We have added information regarding LC-MS and DecyderMS software to the text (page 3, lines 139-147 and page 4, lines 157-169). Due to the short amino acid sequences (as few as five residues), DecyderMS may result in a high number of protein matches.
“MS/MS data-dependent acquisition was enabled. The ion trap was configured for positive ion mode utilizing the manufacturer-specified standard enhanced mode with a scan rate of 8,100 m/z/s. The scan range spanned from m/z 400 to 1,500 for MS, averaging five spectra, and accumulated either 100,000 charges [by ion charge control (ICC)] or for 200 ms, whichever came first. Collision-induced dissociation (CID) fragmentation was performed on the five most intense ions within the m/z range of 200–2,800. The threshold for precursor ion selection was set at an absolute intensity of 5,000. To prevent redundancy, strict active exclusion was applied: a precursor ion was excluded after two spectra and then reintroduced after a brief 0.3-minute interval. The samples were run in three biological replicates of KKU-213A, KKU-213FR, and KKU-213GR, with each replicate undergoing LC-MS analysis in triplicate. A spike of digested BSA was used as an internal standard. The DecyderMS software was employed to quantify peptides in individual samples from MS/MS data [19, 20]. It provides novel 2D and 3D visualizations of LC-MS data to allow for raw data quality assessment and interactive confirmation of results obtained through automated methods for peptide detection, charge state assignments, and peptide matching across multiple LC-MS experiments. Univariate statistical tools such as Student's t-test and ANOVA are available to identify significantly varying peptides among different groups of samples. Briefly, the raw LC-MS/MS data files were imported into the program, which transformed the information into a virtual peptide image. These images were then processed within the Pepdetect module of DecyderMS. The program initially identifies individual peptides by placing a box around them. Subsequently, ion counts for each peptide are integrated, and the Log2 of this number is reported. The process is repeated for each replicate sample. The resulting data are then uploaded to the Pepmatch module, where peptides within different virtual images are matched based on m/z and retention time. Notably, because the data were acquired in an ion trap mass spectrometer, tandem MS ensures accurate peptide matching (i.e., MS/MS spectra consistency). Within the Pepmatch module, each virtual image is assigned to a specific group (KKU-213A, KKU-213FR, and KKU-213GR). Group-to-group comparisons are performed. Integrated peptide counts serve as the basis for statistical analyses, assessing the probability that observed changes are not due to chance. False discovery rates are determined from the data.”
Data Provided:
1. A table listing all identified proteins (File name: RawData_all_proteins).
2. A list of filtered regulated proteins, including 80 upregulated and one downregulated protein (File name: RawData_DEPs).
Explanation for Volcano Plots:
In our opinion, the reason that proteins in the volcano plots are confined to several small areas may be due to the fact that several proteins were very low-abundant and not detected in KKU-213A but could be detected or highly expressed in KKU-213A-FR/KKU-213A-GR cells. Conversely, fewer proteins were not detected in drug-resistant cells but could be detected in the parental cells (KKU-213A). When calculating the fold change, it can be very high, leading to significant differences that cause the observed confinement. If we removed the proteins not detected in KKU-213A but highly expressed in the drug-resistant cells from the analysis, the distribution of the volcano plot could be similar to traditional volcano plots. However, we decided not to exclude those proteins from our analysis, and for this reason, we did not use the volcano plot to represent our data.
Regarding the observation that there should be more than 80 differentially expressed proteins (DEPs) based on the volcano plots, the reviewer is correct. There were indeed more than 80 DEPs. Specifically, there were 137 upregulated proteins and 142 downregulated proteins in KKU-213A vs. KKU-213A-FR, and 112 upregulated proteins and 15 downregulated proteins in KKU-213A vs. KKU-213A-GR. However, we focused on identifying the DEPs that were shared between both KKU-213A-FR and KKU-213A-GR for further analysis, as shown in Figures 2C and 2D. The overlapping DEPs consisted of 80 upregulated proteins and only one downregulated protein.
For clarity, we have added this information to the results section on page 7, lines 287-291.
“In KKU-213A vs. KKU-213FR, we identified a total of 279 DEPs, including 137 upregulated and 142 downregulated proteins. In KKU-213A vs. KKU-213GR, we identified 127 DEPs, comprised of 112 upregulated and 15 downregulated proteins. To identify important genes and pathways in these drug-resistant cells, we searched for overlapping DEPs and found 81 DEPs shared by both KKU-213FR and KKU-213GR, with 80 proteins upregulated (Figure 2c) and only 1 protein downregulated (Figure 2d). The log2 expression values of these proteins in KKU-213A, KKU-213FR, and KKU-213GR were visualized using a heatmap (Figure 2e).”
We have provided the data source for the volcano plots in Excel format as part of the non-published materials. The file name is Log2FC_volcano_plots.